# Adolescent Experiences of the COVID-19 Pandemic and School Closures and Implications for Mental Health, Peer Relationships and Learning: A Qualitative Study in South-West England

**DOI:** 10.3390/ijerph19127163

**Published:** 2022-06-10

**Authors:** Emily Widnall, Emma A. Adams, Ruth Plackett, Lizzy Winstone, Claire M. A. Haworth, Becky Mars, Judi Kidger

**Affiliations:** 1Population Health Sciences, Bristol Medical School, University of Bristol, Bristol BS8 2PS, UK; lizzy.winstone@bristol.ac.uk (L.W.); becky.mars@bristol.ac.uk (B.M.); judi.kidger@bristol.ac.uk (J.K.); 2Population Health Sciences Institute, Newcastle University, Newcastle upon Tyne NE3 4ES, UK; emma.adams@newcastle.ac.uk; 3Division of Primary Care and Population Health, University College London, London NW3 2PF, UK; ruth.plackett.15@ucl.ac.uk; 4School of Psychological Science, University of Bristol, Bristol BS8 1TU, UK; claire.haworth@bristol.ac.uk

**Keywords:** adolescence, peer relationships, mental health, qualitative methods, COVID-19, school closures

## Abstract

The COVID-19 ‘lockdown’ and multiple school closures disrupted the daily lives and routines of the entire UK population. However, adolescents were likely particularly impacted by such measures due to this time being key for social and educational development. This qualitative study explored young people’s experiences of lockdowns and school closures. Fifteen secondary schools within south-west England were initially contacted and three schools participated in recruitment efforts. From December 2020 to March 2021, 25 students aged 14–15 participated in a combination of individual interviews (*n* = 5) and focus groups (*n* = 3). Findings revealed diverse experiences of the pandemic and highlighted the complexity of experiences according to individual student contexts. Three main themes were identified: (1) Learning environments; (2) Connection to peers; (3) Transition, adaptation and coping. These findings highlight the value young people place on face-to-face social contact with close friends, and the sense of structure provided by school, with implications for future home-based learning. Further in-depth qualitative research is needed to continue to understand the varied experiences during the course of the pandemic, particularly longer-term impacts on mental health and learning.

## 1. Introduction

Pandemics have been found to be associated with social isolation and loneliness, disruption to routine and lifestyle as well as feelings of uncertainty and loss of control [1]. COVID-19 is reported to have had a similar detrimental effect on mental health and wellbeing on the general population [2]. Social restrictions are likely to have presented particular challenges to adolescents due to school closures, social distancing requirements and loss of face-to-face social relationships [3,4]. This is of concern as adolescence is an important period for social interaction [5] and adolescents have an increased need for peer connection [6]. Furthermore, this developmental period is associated with an increased incidence of a range of mental health problems, including depression, anxiety and self-harm [6]. In addition to social and mental health impacts, there is added concern around loss of learning due to school closures causing a significant disruption to the education system [7].

### 1.1. Mental Health

The mental health impacts of the COVID-19 pandemic and associated restrictions are still an emerging area of research, particularly for young people. Existing studies provide a mixed picture and there is a lack of pre-pandemic data available to understand change in mental health and well-being over time. These varied results range from a rise in depression and anxiety outcomes [8,9,10], no changes in mental health outcomes [11], and reductions in mental health outcomes, specifically in young people who had higher levels of mental health problems before the pandemic [12]. Our recent longitudinal study, with available pre-pandemic mental health and well-being data to explore change over time, found overall reductions in anxiety during lockdown but a subsequent increase in anxiety on return to school, particularly for adolescents reporting low levels of school connectedness pre-pandemic [13].

Current qualitative evidence assessing COVID-19′s impact on young people’s mental health has largely focused on older adolescents and young adults [14]. A qualitative study of 16–19-year-olds in the UK found heightened emotionality, feelings of loss, change and uncertainty as well as more positive experiences such as recognising the value of self-care and opportunities for relief, growth and development as well as the importance of togetherness [14]. However, less is known about the experiences of mid-adolescents (defined as age 14–17 years). An Irish qualitative study of child and adolescent mental health during the pandemic highlighted experiences of social isolation, depression, and anxiety and increases in maladaptive behaviour among adolescents, but many of these experiences were parent-reported [15].

### 1.2. Peer Relationships

Adolescent social relationships have also been hugely impacted by COVID-19-related public health measures, particularly as school is such a central social context for engaging with peers [16]. During lockdown, adolescents may have spent more time with family and less time with friends, the opposite primary socialisation pattern to early-mid-adolescent norms [17,18]. However, many adolescents are likely to have spent time connecting with friends virtually during the pandemic. A Canadian study of 555 adolescents (mean age 16.2) found their particular concerns centred around schoolwork and peer relationships but, interestingly, those who spent more time on social media reported higher levels of loneliness and depression [19]. This study recommended the importance of monitoring the supportiveness of online relationships. A recently published qualitative study of Canadian 67 children and adolescents (5–14 years) revealed three themes: (1) the irreplaceable nature of friendship; (2) the unsuspected benefits of school for socialisation and (3) the limits and possibilities of virtual socialisation [20]. This study covered a very wide range of ages and interviews took place fairly early on in the pandemic (May–June 2020). One qualitative study carried out in north-west England between September and December 2020 (The ALICE Study) explored early adolescents’ (age 11–14) experiences of the pandemic, resilience processes and self-care strategies [21]. Key issues were focused around missing their peers and face-to-face support from their teachers.

### 1.3. Learning

As well as mental health and social impacts, the COVID-19 pandemic has created huge disruptions to traditional educational practices, and the reopening of schools led to new challenges with many new measures and practices introduced. A recent primary school study in the Netherlands (*n* = 350,000) found little or no progress was made while learning from home, and learning loss was most pronounced for disadvantaged homes [22]. A recent systematic review also revealed that seven out of eight published studies reported learning loss, whilst only one of the seven found instances of learning gains in a particular subgroup [7]. School closures and subsequent changes in school practice have highlighted a number of challenges particularly with regard to online learning such as accessibility, affordability and inclusivity; for example, students without access to technology or those with special educational needs and disabilities [23]. However, several opportunities have also been created as a result of school closures. These include reports of strengthened connections between teachers and parents and new opportunities to teach and learn in innovative ways [24]; for example, a rise in the use of virtual learning environments (e.g., Google Classroom) that provide additional resources and opportunities for learning catch-up at home.

Despite a number of quantitative studies assessing the impact of COVID-19 on adolescent mental health, peer relationships and learning, there remains substantially less qualitative research to support our understanding from a young person’s perspective. Qualitative data are likely to be particularly valuable in helping make sense of the heterogeneity in previous quantitative findings, particularly the varied findings around mental health and well-being.

This is the first study to purposively sample schools in areas of deprivation to capture the voices of students from lower socioeconomic backgrounds who are likely to have been underrepresented in other work. This study also captures young people’s views across multiple phases of the pandemic and extends previous work by including students who were learning at home, remaining in school, or a combination of both.

The aim of the study was to explore the experiences of COVID-19 lockdown school closures and returning to school among adolescents in the south-west of England, with a focus on their mental health and well-being, peer relationships and learning.

## 2. Materials and Methods

### 2.1. Design, Participants and Recruitment

This study aimed to recruit schools in areas of deprivation to address knowledge gaps from convenience survey samples during the pandemic [25,26,27]. A total of 15 schools were contacted by email, 5 schools expressed an interest and 3 schools participated. The schools were purposively recruited based on levels of deprivation, measured by the proportion of students receiving free school meals. The three participating schools had an average of 44.8% of students in receipt of free school meals (at any time during the past six years), which is above the national average in England (27.7%).

This study recruited Year 10 students (aged 14–15 years) as it set out to further explore findings from a previous quantitative survey which found decreases in anxiety during lockdown amongst 13–14-year-olds (13). Within each of the three schools, teachers were asked to invite Year 10 students to express an interest in either a focus group or an individual interview. EW asked teachers to recruit an equal number of boys and girls where possible and students with varying experiences during the pandemic, including those who had been distance learning during both lockdowns, in school for both lockdowns and a combination of distance learning and school learning across the two lockdowns. The total sample consisted of 25 students (14 female and 11 male). No further demographic information was collected. Figure 1 illustrates a summary timeline of school closures and national lockdowns.

### 2.2. Data Collection

Due to ongoing restrictions and pandemic-related measures in place within schools at the time of data collection, flexible recruitment approaches were required. Schools were therefore offered the following formats: (1) focus groups in school, (2) online focus groups, and (3) individual telephone interviews. Student data were therefore collected via a combination of individual semi-structured interviews and focus groups (both online and in school). Individual interviews allowed detailed accounts of participants’ thoughts [28], whilst focus groups allowed interaction data resulting from discussion amongst participants [29,30], particularly to accentuate differences in adolescents’ experiences of school closures and the return to school. A semi-structured approach was used as this provides a flexible and relatively open framework in which participants can talk openly about broad and general questions or topics [31]. The same topic guide was followed for both individual interviews and focus groups and covered the following areas: experiences of lockdown; experiences of returning to school; impact on mental health and well-being; impact on social relationships; experience of change in school year group. Prior to data collection beginning, the topic guide was developed with input from adolescents via a Young Person’s Advisory Group. Table 1 presents a breakdown of the data collection method by gender.

Prior to participating, students were given an information sheet and had an opportunity to ask any questions. All students provided verbal or written consent. All focus groups and interviews were conducted by EW and were audio-recorded. Interviews and focus groups ranged from 20 min to 60 min in length. Individuals received a GBP 10 voucher as a thank you for participating.

Ethical approval was obtained from the University of Bristol’s Faculty of Health Sciences Research Committee (Ref: 2020-7700-7647). Due to the age of the students participating (14–15 years), parental consent was not required or sought (lower age limit of 13 years, DPA 2018), but students and schools were encouraged to share the information sheet with parents or guardians.

### 2.3. Data Analysis

Data were analysed through thematic analysis using the 7-stage framework approach [32]. Audio recordings were transcribed verbatim, reviewed, and checked for accuracy by EW prior to analysis (stage 1). All transcripts were initially read by EW to gain familiarity with the data. To continue familiarisation with the data, two researchers (EW and RP) independently read and annotated four transcripts. RP and EW then independently applied paraphrases or labels to the transcripts to generate and initial list of codes (stage 3). RP and EW then met regularly to discuss and compare these initial codes and agree on a final set of codes to apply to all subsequent transcripts in order to create the analytical framework (stage 4). Although transcripts from differing data collection formats (interview, face-to-face focus group or online focus group) were coded separately to detect any possible differences, no variations in codes were identified by the researchers and, therefore, all transcripts were coded using the same analytical framework. Subsequent transcripts were single-coded by EW applying the agreed analytical framework (stage 5), with further discussion to clarify or expand the framework as needed. EW then charted the data into the framework matrix by creating summaries and identifying key quotes for each category (stage 6). Two authors (EW and EA) then met regularly to interpret the data, mapping connections between categories, identifying central characteristics and comparing data categories between and within cases to generate a set of themes and subthemes (stage 7). Themes and subthemes were then discussed, revised and agreed by all co-authors.

NVivo QSR International version 12 software aided data management and analysis. Codes were both deductive (generated from our topic guide and research questions) and inductive (generated from interview and focus group data). The Framework Method was chosen due to its flexible approach in relation to qualitative health research, particularly due to the complex and individual pandemic-related experiences described by the young people in this study. Charting ensures that researchers are able to pay close attention to describing the data of each student before summarising and interpreting. The Framework Method is not aligned with a particular epistemological viewpoint or theoretical approach which was appropriate considering the novel experience of school closures and exploratory nature of this qualitative study.

As the recruitment approach was the same regardless of interview or focus group format and the same topic guide was used, all transcripts were coded using the same analytical framework. Data collection type was retained in NVivo and has been detailed on the quotes provided in this paper but no thematic differences between formats were identified by the researchers. Previous research has also highlighted that triangulation of qualitative methods (combining individual interviews and focus group data in this case) can enhance data richness [33].

## 3. Results

Experiences of the COVID-19 pandemic, school closures and returning to school were complex and varied between individual students. Students learned through a variety of mediums: some learned at home during both lockdowns, others varied between home learning and in-school learning, and a few remained in school through both lockdowns. The results reflect their collective experiences, and three main themes were identified: (1) Learning environments; (2) Connection to peers; and (3) Transition, adaptation, and coping. Table 2 presents an overview of the main themes and subthemes.

### 3.1. Theme 1: Learning Environments

Adolescents described a wide variation in experiences, all linked to their different environments and learning spaces.

#### 3.1.1. Subtheme 1.1: Challenges of the School Environment

Young people described a sense of relief from being away from some aspects of the in-person school environment during the pandemic. Many students described disliking school and found being a teenager difficult due to challenging dynamics with peers. Bullying was frequently witnessed or experienced.


*“I think I was a little bit relieved to get out such a toxic environment with people I did not like.... I think it is the bullying that people do, picking out everything about you, and you being so exposed to everyone and everything you do is so open”.*
(Female student, Individual Interview)


*“School is like generally a horrible environment...like half the people in school are absolutely horrible and your surrounded by them all day, every day and it’s just constant and if you’re sat next to them on a seating plan it’s just horrible. It’s not the school that’s the problem, it’s just some people and their mindset”*
(Male Student, Focus Group)

Challenges of the school environment related mainly to negative social interactions with peers. Several students also described feelings of excitement and relaxation around the first lockdown, but then an increase in anxiety as the return to school approached.


*“I was more relaxed and excited for the first lockdown and then the closer it is to opening or reopening schools, my anxiety starts to go up”.*
(Female, individual interview)

Another challenging aspect of the school environment referred to distractions within the classroom that negatively impact young people’s learning; for example, peers ‘acting up’ in class and taking attention away from the lesson. Several of the students described the ability to concentrate better on work whilst at home as there were fewer disruptions. Many students also described feelings of pressure within a classroom setting, particularly when required to answer a question in front of peers.


*“It’s just, there’s not the pressure of being asked and everyone staring at you and looking at you, yeah, I just don’t really like being back in with the pressure of everyone around me”.*
(Male student, Focus Group)

#### 3.1.2. Subtheme 1.2: Learning from Home

Student experiences of learning from home varied greatly due to their learning environment and family set-up. A common experience among all three schools was lessons taking place via online platforms where all students had their camera turned off with their microphones on mute, which students found very difficult in terms of engagement. Students that were distance learning at home overwhelmingly were working from their bedrooms, with many not getting out of bed for their school lessons.

*“Nobody puts their camera on or unmutes except the teacher. No one really wants to, I guess...even if the teacher says, ‘Can someone unmute and tell me the answer?’ Everyone just stays silent and maybe types in the chat. Nobody unmutes or turns their camera on”*.(Male student, individual interview)

Some students described wanting to keep their home environments private and not letting others into this space by putting their camera on.


*“Yes, I would not put it on, personally, it is like a safe space and I do not want people in it”.*
(Female student, individual interview)

The majority of student study spaces were in either their own bedroom or a bedroom shared with a sibling, a particular challenge for students in terms of accessing a quiet study space. Students also discussed often staying up late either gaming or on social media due to not having to get up to go to school, and often then spending the majority of the day in their bedroom and using their laptops in bed. Many students struggled with the lack of routine and structure to their day and having no differentiation between school life and home life.


*“At home, I’d do work, but then you keep pushing your work later and later and I ended up doing loads of work at night. Whereas when I was in school, I’d go to school, do my work and have nothing else to do when I got home”.*
(Male student, focus group)


*“Time and days just all blurred into one and there was no routine really”.*
(Female student, focus group)

By contrast, some young people thrived working from home, provided they had a quiet space to learn. They found the lack of disruptions from other students in their typical classes at school beneficial.


*“I found myself doing more work than I do in school normally. Because I didn’t have people distracting me”*
(Female student, individual interview)

### 3.2. Theme 2: Connection to Peers

Throughout interviews and focus groups, peer connections and social relationships were a central theme that impacted many student experiences.

#### 3.2.1. Subtheme 2.1: The Importance of Peers

Adolescents described the importance of being around people their own age in terms of their personal development.


*“We’re growing, we need our peers to check ourselves against. It’s part of growing really. To me having people, your friends and your peers, with you so you can change”.*


Several young people reported missing in-person social interactions from being away from school and, due to social distancing guidelines and restrictions, students who were still attending school also struggled being away from friends and described feelings of loneliness while at school because not everyone was attending.


*“I’ve begged my friends to come into school if they can. ...It’s still social interaction but it’s not with the people that I get happiness from. ...So it’s still, I guess, a bit lonely”.*
(Female, individual interview)

Students reflected on changes in friendships, which included losing friends, making new friends and realising the importance of ‘*true friendships*’. Students who remained in school during school closures also discussed opportunities to chat to new peers and make new friends.


*“I preferred being in school with just a small selection of people, compared to having loads of people in, because you kind of get to know people that you never really talk to...then as soon as everybody comes back to school, everything changes, because you go back to your old friend groups and you don’t really talk to each other anymore”.*
(Female student, individual interview)

Some young people described feeling concerned for their friends who had gone quiet during lockdown.


*“I’ve seen a lot of people, people that I know, some of them have gone really antisocial. Not in contact with many people at all”.*
(Female student, individual interview)

Students described the process of checking in on friends and sometimes not experiencing this in return. Others appreciated their friends who were supportive throughout the pandemic.


*“I can say, from my experience, I’ve fallen out with loads of my friends, because it’s me doing most of the chasing, to make sure people are okay. And that is just not really how friendships are supposed to work... Most of my friendships are just gone. But I’m thankful for COVID that I could see that they weren’t really friends, if that makes sense... And you can see which ones are actually there for you and would check on you and make sure you’re okay”.*
(Female student, individual interview)

#### 3.2.2. Subtheme 2.2: Online vs. Face-to-Face Socialising

Students made frequent comparisons to socialising online compared to seeing their friends in-person both in and out of school. Many described connecting virtually with friends via social media, but perceived this to be inferior to face-to-face contact.


*“I mean I did want to speak to people but I’d rather meet people face-to-face and enjoy time with them but- Yes, I did try to speak to people but, again, they have their own mental health and stuff going on. Sometimes it’s not as easy as just you both being in the right frame of mind to call your friend or be texting and be replying quickly and, you know, those kinds of things”.*
(Male student, individual interview)


*“I think it is not being able to have in person interaction with people, and to just tell them how you are doing and like how you are feeling...but social media is a big help in terms of reaching out to people and keeping connected”.*
(Female student, individual interview)

Some students said they found keeping in touch with friends easy due to use of social media but still lacked face-to-face interaction:


*“I think it is pretty easy because there are only four of us in my friendship group, and so we do group FaceTimes at least once a week, if not every day. I did not really mind it because we use social media a lot between us lot. But it was a bit different not going out on the weekends and seeing them”.*
(Female student, individual interview)

Although young people discussed the positives of social media for keeping in touch with peers, excessive screen time was a common concern. Students were using laptops all day for home learning as well as prolonged screentime for socialising, gaming and checking social media. Young people discussed the importance of screen breaks due to spending so much time online.


*“My screen time went up massively, I was doing my lessons on my laptop and then also on my phone. So my breaks from my laptop were on my phone so screen time went up loads”*
(Male student, focus group)

### 3.3. Theme 3: Transitions, Adaptation and Coping

#### 3.3.1. Subtheme 3.1: Transition Back to School

All students agreed they were looking forward to returning to school, despite the ongoing challenges of the school environment described above. Young people were most positive about reintroducing routine and structure to their days, and being able to see friends again. However, this was also accompanied by feelings of stress and anxiety around adapting to a return to busy social environments.


*“I’m pretty stressed but more excited to see everyone”*
(Male student, focus group)


*“I think I am a bit relieved to be back and having the social aspect, then it is still a bit nerve-wracking being around people that do not particularly like me, and I do not particularly like them”.*
(Female, Individual interview)

After returning to school, students described how school felt very different with a combination of safety measures and ongoing restrictions such as year group bubbles (students only being able to interact with students in their year group).

*“It kind of feels like you are not meant to be here. It feels wrong because you haven’t been in there for so long”*. (Female, individual interview)

As well as transitioning between school and home, and between changing social distancing measures, this cohort of students also faced the added complexity of a year group transition, which was a central area of concern. Students felt they had lost a year of learning, which left them underprepared for an important year of school where they would face exams and increased academic pressure.


*“Stressful, Year 10 is where you learn everything for your exams and then Year 11 is mainly revising. And if we miss out and don’t learn everything this year, next year is going to be so so hard. But then we also have no motivation to catch up at home”.*
(Male student, focus group)

#### 3.3.2. Subtheme 3.2: Uncertainty and Change

As well as the increased stress and anxiety around returning to a new year group at school, there was concern around expectations and continued uncertainty around changes to exam procedures.


*“I am finding it really hard to cope because it’s really overwhelming thinking about exams and how I’ve gotten nowhere and no ambitions of what I want in the future”.*
(Female student, focus group)


*“I don’t really know if exams are still happening? We don’t know what’s going on, we just don’t know 100% and if they’re going to get cancelled. We don’t know whether we’re doing full papers, or predicted grades. That’s more stressful, because we have to do all of the work in such a constricted time”*
(Male student, focus group)

Students discussed the difficulties of constantly having to adapt and adjust to sudden changes, especially short-term returns to home-based learning if an outbreak emerged in the school.


*“Adapting, like when you adapt to something and it suddenly changes then you have to adapt again, it keeps changing, its constant change and it impacts our learning”.*
(Female student, Focus Group)


*“It’s the adjusting, because it’s like the build up of all the stress about going back to school and then you get back and it’s OK and then you’re sent off again and you build yourself up again and you’re back and no one acknowledges it”.*
(Male student, Focus Group)

#### 3.3.3. Subtheme 3.3: Help-Seeking and Coping Strategies

Young people discussed the importance of help-seeking and talking about their feelings throughout the pandemic. Many students agreed that help-seeking in school was difficult, particularly not wishing to discuss bullying and challenging peer relationships with teachers.

*“I’d say 90% of the time no one wants to tell the teacher... Because nothing gets done about it in school. Most of the time they just push it to the side or get someone to apologise, but it doesn’t fix the main problem”*.(Male student, focus group)

Students felt stigma around help-seeking was still present and expressed concern about the lack of support available for those who request it.


*“They’d [teachers] probably sit you down, talk about it and be like OK cool, just for probably like 10 min, and that’s it, but then they won’t like give you extra support for it”.*
(Female, focus group)

During the interviews, students also reflected on what may be beneficial to other students their age and offered advice to other peers on how they had coped during lockdown and school closures, as well as ideas for how students may settle back into school. Several students discussed the importance of downtime away from lessons and formal learning as well as the need for some alone time when adapting to being back in a busy school environment.


*“I think sometimes they just need some downtime from a lesson. Like when we had our screen breaks after the lesson, I think that was like really important for people to just go outside and have a breath of fresh air, and to just be by themselves. I think alone time is a really big one for being back in such a big environment with people”.*
(Female student, individual interview)

## 4. Discussion

The COVID-19 pandemic led to drastic changes in schooling within England. School closures have offered an opportunity for reflection on the current education system and the multifaceted pressures placed on adolescents. This study aimed to explore adolescents’ experiences during the COVID-19 lockdowns, specifically the psychological and social impacts on this age group. This research used an inductive qualitative research design. Although there were individual variations in experience, our findings suggest that young people generally adapted well to school closures and home learning, but overwhelmingly missed in-person social contact with friends both at school and in their personal time. However, face-to-face social interactions (such as bullying and difficult peer relationships) were also identified as common challenges of the in-person school environment for young people. Young people’s willingness to return to school varied based on their connection to school and relationships with peers and teachers, but there was a positive consensus regarding the importance of returning to school in order to socialise with close friends face-to-face and get back to routines.

Many young people experienced a sense of relief during the first lockdown due to being away from the in-person school environment. This may, in part, explain the reduction in anxiety reported by adolescents in our recent survey study [13]. Existing research demonstrates the associations between the school environment and mental health in adolescence [34]. Our findings document concerns that young people have around the school environment, class sizes, pressure from teachers, and bullying. Several students discussed peer group numbers feeling overwhelming in school; therefore, one potential implication of these results could be to reduce class sizes. The findings concerning challenges of the school environment suggest the need for peer relationships and bullying be dealt with more effectively in schools to help increase young people’s connection to school as well as a sense of belonging and safety in school. Our findings also revealed the importance of learning spaces, with a distinction between a safe, private home environment, and an open, exposed school environment surrounded by a large number of peers.

Experiences of online learning also varied greatly with some students struggling with motivation whilst others finding they were getting more work done than when in school. Past research has found that students who are more successful in online learning tend to prefer independent learning, have higher intrinsic motivation and have greater time management, literacy, and technology skills [35]. It is therefore likely that some students will have experienced a greater loss in learning than others which should be carefully considered as schools continue with academic catch-up. Existing research during the pandemic has also demonstrated that maintaining relationships during online learning is a significant challenge and a lack of togetherness with peers has been found to be one of the most critical factors [36] which was reflected throughout discussions with students.

The importance of peer relationships throughout adolescence is well documented in the literature [37,38]. In line with recent findings from a Canadian study [20], our results confirm that the primary difficulty for young people during lockdown and school closures was not spending time with their friends. This study highlights not only the importance of peer relationships to early adolescents, but the value young people place specifically on face-to-face contact. Again, a finding consistent with results from the Canadian sample shows that although virtual communication is convenient, it does not replace face-to-face peer interactions [20]. This lack of social contact was described by adolescents despite regular online contact with friends, with some students discussing feeling lonely and noticing that their friends had gone very quiet and not kept in touch. The findings highlight the need to maximise opportunities to enhance in-person peer relationship development given the lack of face-to-face contact young people experienced during lockdown. Particularly given recent findings that feelings of loneliness have increased since the start of the pandemic [39] and the impacts of social isolation and loneliness on adolescent mental health [40].

Although recent findings from the ALICE study have highlighted the ability for early adolescents to be resilient and adapt [21], adolescents in this study discussed how challenging they found ongoing uncertainty and constant changes in social distancing rules as well as last minute announcements of school closures and return to distance learning. Young people voiced that it was important for there to be a period of stability for them whether that be at home or in school. Students discussed particular frustration with the uncertainty around exams and what format these would take, which echoes findings from a recent qualitative study in the UK in which students (aged 13–17) reported feeling stressed with the lack of clear guidance on how schools would manage coursework and exams amid school closures [41].

Routine and structure were an important implication for teenage mental health amidst ongoing uncertainty throughout the pandemic. The absence of a structured setting of school during the pandemic has been highlighted as an important coping mechanism for young people with mental health issues [42]. This absence of routine was reflected in our findings, particularly around the routine of travelling to and from school and the structure of a school day, suggesting this may have acted as a coping mechanism for some students. This could also be linked with a sense of school belonging, which we know to be of importance for academic and psychological functioning in adolescence [43]. The importance of routine and structure also echoes findings from recent qualitative studies [21,44].

The breadth of experiences documented by the adolescents in this study reveals the importance of individual preferences and contexts in assessing the mental health impacts of the COVID-19 pandemic. It is likely that the vast heterogeneity in experiences has increased existing health inequalities. Careful consideration is needed to ensure interventions meet the needs of a wide range of experiences and outcomes, likely by embedding whole-school approaches. Furthermore, our findings highlight the importance of involving young people in school reform beyond the pandemic, empowering the student voice to ensure responses are acceptable and appropriate for this population. Our study provides important implications for future distance learning if there is another lockdown, particularly around encouraging students to create routine and structure to their day and providing guidance around positive learning environments.

Mental health implications predominantly centred around anxiety about returning to school, illustrating a clear need for additional support for students as they continue to adjust being back at school amidst the ongoing pandemic. Findings relating to peer connectedness also highlighted the importance of young people having a core group of friends who they could rely on and talk to about their feelings. The pandemic appears to have given adolescents a chance to reflect on and assess what is really meaningful to them, which aligns with a qualitative study which found increased self-awareness amongst young people [41]. Our findings particularly noted the role of the pandemic in understanding the importance of relationships—aligning with a recent Portuguese adolescent study [3]. As many students discussed missing face-to-face interaction with peers, it is likely that adolescents will need a period of social catch-up as well as academic catch-up. Future work should look to further understand the support students received on return to school and longer-term mental health outcomes for teenagers as the pandemic continues. Our findings on young people’s apprehension around seeking help from teachers suggest that further efforts could be focused on help-seeking and open conversations about mental health in schools, which in turn may have positive implications for young people’s connectedness to school.

### Limitations

This study involved twenty-five 14–15-year-old students participating from three schools in south-west England. Study findings therefore may not be transferable to older adolescents or to younger adolescents experiencing different transitions, e.g., those just starting secondary school. Due to localised recruitment, findings may also be specific to the region given the differing rates of infection across England. Due to sampling by deprivation, some issues (such as needing to work in a bedroom or a room shared with siblings) may be less of an issue in a high-income sample. However, it is important to note that although the schools were in deprived areas, it was not deemed appropriate to sample pupils on this basis. As teachers were asked to coordinate student recruitment, it is also possible that teachers may have deliberately selected students who were doing particularly well, or who were struggling. Given the complexity and individual nature of young people’s experiences of the pandemic, future research could look to using more individual styles of analysis, for example, interpretative phenomenological analysis to gain further insight into the context surrounding young people’s experiences. Given the evolving and continuing nature of the pandemic, future research would also benefit from longitudinal qualitative methods to capture young people’s perspectives across multiple time points.

## 5. Conclusions

This study provides important insights into adolescent experiences of COVID-19 and highlights the value that young people place on face-to-face social contact. The study also brings attention to students’ concerns with the school environment, particularly in relation to bullying and challenging peer relationships. These findings suggest that improving school connectedness and peer relationships within school are two areas of priority and potential areas of intervention.

## Figures and Tables

**Figure 1 ijerph-19-07163-f001:**
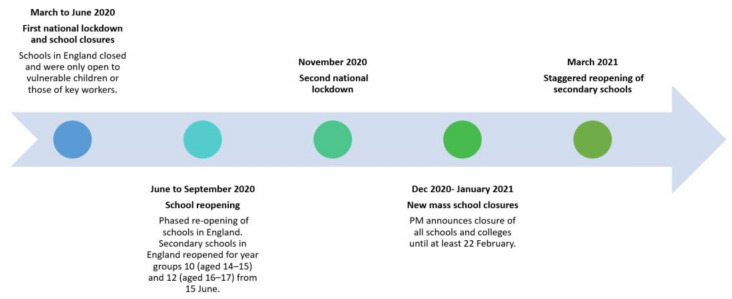
Timeline of school closures and national lockdowns.

**Table 1 ijerph-19-07163-t001:** Participant and data collection summary.

Data Collection Category	Female	Male
1:1 interview	4	1
Online focus group	6	6
In school focus group	4	4
**Total**	**14**	**11**

**Table 2 ijerph-19-07163-t002:** Overview of themes and subthemes.

Theme	Subthemes
1. Learning environments	1.1 Challenges of the school environment
	1.2 Learning from home
2. Connection to peers	2.1 The importance of peers
	2.2 Online vs. face-to-face contact
3. Transition, adaptation and coping	3.1 Transition back to school
	3.2 Uncertainty and change
	3.3 Help-seeking and coping strategies

## Data Availability

Summaries of the data presented in the study are available on request from the corresponding author. The data are not publicly available due to containing potentially identifiable information about participants.

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
