# Peer review of "Adolescent Experiences of the COVID-19 Pandemic and School Closures and Implications for Mental Health, Peer Relationships and Learning: A Qualitative Study in South-West England"

_ijerph, 2022, doi:10.3390/ijerph19127163_

Round 1
Reviewer 1 Report
This is a well described research report and is quite informing of adolescents experiences in these times of COVID-19 pandemic.
I would encourage the authors to provide descriptions of any additional strategies of rigor they employed in addition to inter-rater reliability strategies they employed in developing the coding system as they analyzed the transcripts.
Can the authors provide more details about the focus group format versus the individual interviews in conducting the analysis and combining them?; how they treated the focus group responses versus individual interviews in the analysis? the rationale for combining them in the analysis?
In conducting thematic analysis, did the authors follow any particular-qualitative research/researcher tradition driven approach? If so, it would be helpful to indicate that.
Author Response
Reviewer 1, Comment 1: This is a well described research report and is quite informing of adolescents experiences in these times of COVID-19 pandemic.
Authors response: The authors would like to thank the reviewer for their positive comments about the study.
Reviewer 1, Comment 2: I would encourage the authors to provide descriptions of any additional strategies of rigor they employed in addition to inter-rater reliability strategies they employed in developing the coding system as they analyzed the transcripts.
Authors response: The authors thank the reviewer for raising this point. We have revised the methods to provide a more detailed description of the framework approach used for the qualitative analysis. This approach does not adopt a double coding strategy or formal testing of inter-rater reliability, however two researchers worked closely together to create a comprehensive analytic framework to code each transcript and chart data into a coding matrix.
Changes to the manuscript:
Data were analysed through thematic analysis using the 7 stage framework approach[19]. Audio recordings were transcribed verbatim, reviewed, and checked for accuracy by EW prior to analysis (stage 1). All transcripts were initially read by EW to gain familiarity with the data. To continue familiarisisation with the data, two researchers (EW and RP) independently read and annotated four transcripts. RP and EW then independently applied paraphrases or labels to the transcripts to generate and initial list of codes (stage 3). RP and EW then met regularly to discuss and compare these initial codes and agree on a final set of codes to apply to all subsequent transcripts in order to create the analytic framework (stage 4). Subsequent transcripts were single-coded by EW applying the agreed analytical framework (stage 5), with further discussion to clarify or expand the framework as needed. EW then charted the data into the framework matrix (stage 6). Two authors (EW and EA) then met regularly to interpret the data, mapping connections between categories, identifying central characteristics and comparing data categories between and within cases to generate a set of themes and sub-themes (stage 7). Themes and sub-themes were then discussed, revised and agreed by all co-authors.
NVivo version 12 software aided data management and analysis. Codes were both deductive (generated from our topic guide and research questions) and inductive (generated from interview and focus group data). The Framework Method was chosen due to its flexible approach in relation to qualitative health research, particularly due to the complex and individual pandemic-related experiences described by the young people in this study. Charting ensures that researchers are able to pay close attention to describing the data of each student before summarizing and interpreting. The Framework Method is not aligned with a particular epistemological viewpoint or theoretical approach which was appropriate considering the novel experience of school closures and exploratory nature of this qualitative study.
Reviewer 1, Comment 3: Can the authors provide more details about the focus group format versus the individual interviews in conducting the analysis and combining them?; how they treated the focus group responses versus individual interviews in the analysis? the rationale for combining them in the analysis?
Authors response:
The authors thank the author for identifying the need to clarify the decision on combining the different qualitative methods. The varied format of data collection was offered as a way of providing flexibility to schools at a challenging time to participate in research, this has now been made clearer in the manuscript, as well as clarifying that the topic guides for both formats remained the same. Data collection format was classified in NVivo and therefore focus group transcripts and interview transcripts were coded separately, however, after finding no variations in codes between the two formats, all transcripts were coded using the same analytic framework. This has now all been described in the manuscript.
Changes to the manuscript:
“Both individual interviews and focus groups used the same topic guide.”
“Due to ongoing restrictions and pandemic-related measures in place within schools at the time of data collection, flexible recruitment approaches were required. Schools were therefore offered the following formats: 1) focus groups in school 2) online focus groups and 3) individual telephone interviews. Student data was therefore collected via a combination of individual semi-structured interviews and focus groups (both online and in school).”
“As the recruitment approach was the same regardless of interview or focus group format and the same topic guide was used, all transcripts were coded using the same analytic framework. Data collection type was categorised in NVivo and has been detailed on the quotes provided in this paper but no thematic differences between formats were identified by the researchers.”
Reviewer 1, Comment 4: In conducting thematic analysis, did the authors follow any particular-qualitative research/researcher tradition driven approach? If so, it would be helpful to indicate that.
Authors response:
As discussed within the response to reviewer one’s first comment regarding inter-rater reliability and coding transcripts, a full description of the thematic analysis approach has now been added to the manuscript, including the formal approach used; The Framework Method (Gale et al. 2013, https://bmcmedresmethodol.biomedcentral.com/articles/10.1186/1471-2288-13-117)
Changes to the manuscript:
*same changes as described to Reviewer 1, Comment 2.
Reviewer 2 Report
1. The title focused on the implications of mental health, peer relationships and learning between school opens and closures. But in the section of the introduction, I cannot find the explorations about these issues. I suggest the author should provide more literature analysis about the mental health, peer relationships and learning for adolescent experiences of the Covid-19 pandemic. In addition, the author should analyze the advantages and disadvantages between the school opens and closures.
2. In the section of the materials and methods, I suggest the author should provide more theoretical backgrounds of the tools of the interviews and focus group. The author also explain these tools or items how to assist the author to collect the reliable and valid data to improve the qualitative analysis and exploration of the results.
3. In the table 2 and related statements, I suggest the author should provide more discussion about the organization and framework of the overview of themes and subthemes.
4. In the section of the results, I suggest the author should employ the above theoretical discourse from literature analysis to examine the interrelationships and articulations among the data of the interviews and focus groups.
5. In the section of the discussion, I suggest the author should focus on the researching questions and targets with the meaningful results to construct the useful conclusions and practical suggestion for related institutes.
Author Response
Reviewer 2, Comment 1: The title focused on the implications of mental health, peer relationships and learning between school opens and closures. But in the section of the introduction, I cannot find the explorations about these issues. I suggest the author should provide more literature analysis about the mental health, peer relationships and learning for adolescent experiences of the Covid-19 pandemic. In addition, the author should analyze the advantages and disadvantages between the school opens and closures.
Authors response: The authors agree that the introduction could benefit from some clearer exploration of mental health, peer relationships and learning specifically in the context of Covid-19. We have now reframed the introduction to analyse the existing literature more clearly for each of these three areas and added subtitles within the introduction to help make this clearer.
Additionally, the authors have now added some advantages and disadvantages of school closures with reference to existing literature.
Changes to the manuscript:
Pandemics have been found to be associated with social isolation and loneliness, disruption to routine and lifestyle as well as feelings of uncertainty and loss of control[1]. COVID-19 is reported to have had a similar detrimental effect on mental health and well-being on the general population[2]. Social restrictions are likely to have presented particular challenges to adolescents due to school closures, social distancing requirements and loss of face-to-face social relationships [3, 4]. This is of concern as adolescence is an important period for social interaction [5] and adolescents have an increased need for peer connection [6]. Furthermore, this developmental period is associated with an increased incidence of a range of mental health problems, including depression, anxiety and self-harm [6]. In addition to social and mental health impacts, there is added concern around loss of learning due to school closures causing a significant disruption to the education system[7]
Mental Health
Mental health impacts of the COVID-19 pandemic and associated restrictions is still an emerging area of research, particularly for young people. Existing studies are largely quantitative and provide a mixed picture and there is a lack of pre-pandemic data available to understand change in mental health and well-being over time. These varied results, range from a rise in depression and anxiety outcomes [8-10], no changes in mental health outcomes [11], and reductions in mental health outcomes, specifically in young people who had higher levels of mental health problems before the pandemic [12].Our recent longitudinal study, with available pre-pandemic mental health and well-being data to explore change over time, found overall reductions in anxiety during lockdown but a subsequent increase in anxiety on return to school, particularly for adolescents reporting low levels of school connectedness pre-pandemic[13].
Current qualitative evidence assessing COVID-19’s impact on young people’s mental health has focused on older adolescents and young adults [14], A qualitative study of 16-19 year olds in the UK found heightened emotionality, feelings of loss, change and uncertainty as well as more positive experiences such as recognizing the value of self-care and opportunities for relief, growth and development as well as the importance of togetherness [14]. But less is known about the experiences of mid-adolescents (defined as age 14-17 years). An Irish qualitative study of child and adolescent mental health during the pandemic highlighted experiences of social isolation, depression, anxiety and increases in maladaptive behaviour among children and adolescents [15].
Peer relationships
Adolescent social relationships have also been hugely impacted by COVID-19 related public health measures, particularly as school is such a central social context for engaging with peers[16]. During lockdown, adolescents may have spent more time with family and less time with friends, the opposite primary socialisation pattern to early-mid adolescent norms[17, 18]. However, many adolescents are likely to have spent time connecting with friends virtually during the pandemic. A Canadian study of 555 adolescents (mean age 16.2) found their particular concerns centered around schoolwork and peer relationships but interestingly those who spent more time on social media reported higher levels of loneliness and depression [19]. This study recommended the importance of monitoring the supportiveness of online relationships. A recently published qualitative study of Canadian 67 children and adolescents (5-14 years) revealed three themes: 1) the irreplaceable nature of friendship; 2) the unsuspected benefits of school for socialisation and 3) the limits and possibilities of virtual socialisation[20]. This study covered a very wide range of ages and interviews took place fairly early on in the pandemic (May-June 2020). One qualitative study carried out in North West England between September-December 2020 (The ALICE Study), explored early adolescents’ (age 11-14) experiences of the pandemic, resilience processes and self-care strategies. [21]. Key issues were focused around missing their peers and face-to-face support from their teachers.
Learning
As well as mental health and social impacts, the COVID-19 pandemic has created huge disruptions to traditional educational practices, and the reopening of schools led to new challenges with many new measures and practices introduced. A recent primary school study in the Netherlands (n= 350,000) found little or no progress was made while learning from home, and learning loss was most pronounced for disadvantaged homes[22]. A recent systematic review also revealed that seven out of eight published studies reported learning loss, whilst only one of the seven found instances of learning gains in a particular subgroup[7]. School closures and subsequent changes in school practice have highlighted a number of challenges particularly with regard to online learning such as accessibility, affordability and inclusivity, for example students without access to technology or those with special educational needs and disabilities[23]. However, several opportunities have also been created as a result of school closures. These include reports of strengthened connections between teachers and parents and new opportunities to teach and learn in innovative ways[24], for example a rise in the use of virtual learning environments (e.g. Google Classroom) that provide additional resources and opportunities for learning catch-up at home.
Despite a number of quantitative studies assessing the impact of COVID-19 on adolescent mental health, peer relationships and learning, there remains substantially less qualitative research to support our understanding from a young person’s perspective. Qualitative data is likely to be particularly valuable in helping make sense of the heterogeneity in previous quantitative findings, particularly the varied findings around mental health and well-being.
This is the first study to purposively sample schools in areas of deprivation to capture the voices of students from lower socioeconomic background who are likely to have been underrepresented in other work. This study also captures young people’s views across multiple phases of the pandemic and extends previous work by including students who were learning at home, remaining in school, or a combination of both.
The aim of the study was to explore the experiences of COVID-19 lockdowns, school closures and returning to school among adolescents in the south-west of England with a focus on their mental health and well-being, peer relationships and learning.
Reviewer 2, Comment 2: In the section of the materials and methods, I suggest the author should provide more theoretical backgrounds of the tools of the interviews and focus group. The author also explain these tools or items how to assist the author to collect the reliable and valid data to improve the qualitative analysis and exploration of the results.
Authors response: The authors would like to thank the reviewer for this recommendation. The justification for the use of interviews and focus groups has now been added to the materials and methods section along with theoretical references. Additionally, the authors have added previous literature to justify combining focus group and interview data for a richer analysis (Lambert & Loiselle, 2008). An explanation of the use of a semi-structured approach has also been provided.
Changes to the manuscript:
Due to ongoing restrictions and pandemic-related measures in place within schools at the time of data collection, flexible recruitment approaches were required. Schools were therefore offered the following formats: 1) focus groups in school 2) online focus groups and 3) individual telephone interviews. Student data was therefore collected via a combination of individual semi-structured interviews and focus groups (both online and in school). Individual interviews allowed detailed accounts of participants’ thoughts[28], whilst focus groups allowed interaction data resulting from discussion amongst participants[29, 30], particularly to accentuate differences in adolescents experiences of school closures and the return to school. A semi-structured approach was used as this provides a flexible and relatively open framework in which participants can talk openly to broad and general questions or topics[31]. The same topic guide was followed for both individual interviews and focus groups and covered the following areas: experiences of lockdown; experiences of returning to school; impact on mental health and well-being; impact on social relationships; experience of change in school year group. Prior to data collection beginning, the topic guide was developed with input from adolescents via a Young Person’s Advisory Group. Table 1 presents a breakdown of the data collection method by gender.
Reviewer 2, Comment 3: In the table 2 and related statements, I suggest the author should provide more discussion about the organization and framework of the overview of themes and subthemes.
Authors response: The authors would like to thank the reviewer for this comment. We have revised the data analysis section to provide a thorough breakdown of the stages involved in data analysis and interpretation. This now includes a further detailed discussion about how the overview of themes and subthemes were created using the Framework Approach within the data analysis section.
Changes to the manuscript:
Data were analysed through thematic analysis using the 7 stage framework approach[19]. Audio recordings were transcribed verbatim, reviewed, and checked for accuracy by EW prior to analysis (stage 1). All transcripts were initially read by EW to gain familiarity with the data. To continue familiarisisation with the data, two researchers (EW and RP) independently read and annotated four transcripts. RP and EW then independently applied paraphrases or labels to the transcripts to generate and initial list of codes (stage 3). RP and EW then met regularly to discuss and compare these initial codes and agree on a final set of codes to apply to all subsequent transcripts in order to create the analytic framework (stage 4). Subsequent transcripts were single-coded by EW applying the agreed analytical framework (stage 5), with further discussion to clarify or expand the framework as needed. EW then charted the data into the framework matrix (stage 6). Two authors (EW and EA) then met regularly to interpret the data, mapping connections between categories, identifying central characteristics and comparing data categories between and within cases to generate a set of themes and sub-themes (stage 7). Themes and sub-themes were then discussed, revised and agreed by all co-authors.
NVivo version 12 software aided data management and analysis. Codes were both deductive (generated from our topic guide and research questions) and inductive (generated from interview and focus group data). The Framework Method was chosen due to its flexible approach in relation to qualitative health research, particularly due to the complex and individual pandemic-related experiences described by the young people in this study. Charting ensures that researchers are able to pay close attention to describing the data of each student before summarizing and interpreting. The Framework Method is not aligned with a particular epistemological viewpoint or theoretical approach which was appropriate considering the novel experience of school closures and exploratory nature of this qualitative study.
Reviewer 2, Comment 4: In the section of the results, I suggest the author should employ the above theoretical discourse from literature analysis to examine the interrelationships and articulations among the data of the interviews and focus groups.
Authors response: The authors would like to thank the reviewer for this comment. We have now provided further reference to existing literature within the discussion section when interpreting the results of the study. The discussion section has also been restructured so it addresses the main findings in relation to key literature more clearly. The discussion has also been reordered so it summarises the findings in order of the themes and subthemes in Table 2 which the authors believe will be easier for the reader to follow in relation to the presentation of the results section.
Changes to the manuscript:
Experiences of online learning also varied greatly with some students struggling with motivation whilst others finding they were getting more work done than when in school. Past research has found that students who are more successful in online learning tend to prefer independent learning, have higher intrinsic motivation and have greater time management, literacy, and technology skills [36]. It is therefore likely that some students will have experienced a greater loss in learning than others which should be carefully considered as schools continue with academic catch-up. Existing research during the pandemic has also demonstrated that maintaining relationships during online learning is a significant challenge and a lack of togetherness with peers has been found to be one of the most critical factors [37] which was reflected throughout discussions with students.
The importance of peer relationships throughout adolescence is well documented in the literature [38, 39]. In line with recent findings from a Canadian study [23], our results confirm that the primary difficulty for young people during lockdown and school closures, was not spending time with their friends. This study highlights not only the importance of peer relationships to early adolescents, but the value young people place specifically on face-to-face contact. Again, a finding consistent with results from the Canadian sample showing that although virtual communication is convenient, it does not replace face-to-face peer interactions [23]. This lack of social contact was de-scribed by adolescents despite regular online contact with friends with some students discussed feeling lonely and noticing that their friends had gone very quiet and not kept in touch. The findings highlight the need to maximise opportunities to enhance in-person peer relationship development given the lack of face-to-face contact young people experienced during lockdown.
Although recent findings from the ALICE study have highlighted the ability for early adolescents to be resilient and adapt [21], adolescents in this study discussed how challenging they found ongoing uncertainty and constant changes in social distancing rules as well as last minute announcements of school closures and return to distance learning. Young people voiced that it was important for there to be a period of stability for them whether that be at home or in school. Students discussed particular frustration with the uncertainty around exams and what format these would take which echoes findings from a recent qualitative study in the UK in which students (aged 13-17) re-ported feeling stressed with the lack of clear guidance on how schools would manage coursework and exams amid school closures.
Routine and structure was an important implication for teenage mental health amidst ongoing uncertainty throughout the pandemic. The absence of a structured set-ting of school during the pandemic has been highlighted as an important coping mechanism for young people with mental health issues [41]. This absence of routine was reflected in our findings, particularly around the routine of travelling to and from school and the structure of a school day, suggesting this may have acted as a coping mechanism for some students. This could also be linked with a sense of school belonging, which we know to be of importance for academic and psychological functioning in adolescence [42]. The importance of routine and structure also echoes findings from recent qualitative studies [21, 43].
Mental health implications predominantly centered around anxiety about returning to school illustrating a clear need for additional support for students as they continue to adjust being back at school amidst the ongoing pandemic. Findings relating to peer connectedness also highlighted the importance of young people have a core group of friends who they could rely on and talk to about their feelings. The pandemic appears to have given adolescents a chance to reflect on and assess what is really meaningful to them, which aligns with a qualitative study which found increased self-awareness amongst young people [42]. Our findings particularly noted the role of the pandemic in understanding the importance of relationships – aligning with a recent Portuguese adolescent study [3]. As many students discussed missing face-to-face interaction with peers, it is likely that adolescents will need a period of social catch-up as well as academic catch up. Future work should look to further understand the support students received on return to school and longer term mental health outcomes for teenagers as the pandemic continues. Our findings on young people’s apprehension around seeking help from teachers suggest that further efforts could be focused on help-seeking and open conversations about mental health in schools, which in turn may have positive implications for young people’s connectedness to school.
Reviewer 2, Comment 5: In the section of the discussion, I suggest the author should focus on the researching questions and targets with the meaningful results to construct the useful conclusions and practical suggestion for related institutes.
Authors response: Due to a lack of robust qualitative literature on adolescent experiences of the pandemic and due to national school closures being a novel occurrence with little prior literature, this study was exploratory in nature and we therefore did not set specific research questions. However, as discussed in response to comment 4, a number of changes have been made to the discussion in order to highlight the main conclusions. We also provide the following practical suggestions within the manuscript:
- “The findings highlight the need to maximise opportunities to enhance in-person peer relationship development given the lack of face-to-face contact young people experienced during lockdown.”
- “Several students discussed peer group numbers feeling overwhelming in school, therefore one potential implication of these results could be to reduce class sizes.”
- “Our findings on young people’s apprehension around seeking help from teachers suggest that further efforts could be focused on help-seeking and open conversations about mental health in schools, which in turn may have positive implications for young people’s connectedness to school.”
- “Careful consideration is needed to ensure interventions meet the needs of the wide range of experiences and outcomes, likely by embedding whole school approaches.”
- Furthermore, our findings highlight the importance of involving young people in school reform beyond the pandemic, empowering the student voice to ensure responses are acceptable and appropriate for this population.
- Our study provides important implications for future distance learning if there is another lockdown, particularly around encouraging students to create routine and structure to their day and providing guidance around positive learning environments.
- “Improving school connectedness and peer relationships within school are two areas of priority and potential areas of intervention.”
Round 2
Reviewer 2 Report
The author mostly incorporated my suggestions and revised the manuscript. The manuscript could be published in this form for the journal.